# Stator Core Shape Design for Low Core Loss and High Power Density of a Small Surface-Mounted Permanent Motor

**DOI:** 10.3390/s20051418

**Published:** 2020-03-05

**Authors:** Naoya Soda, Masato Enokizono

**Affiliations:** 1Department of Electrical and Electronic Systems Engineering, Graduate School of Science and Engineering, Ibaraki University; Hitachi 316-8511, Japan; 2Vector Magnetic Characteristic Technical Laboratory, Usa 879-0442, Japan; enoki@oita-u.ac.jp

**Keywords:** core loss, motor core, power density, shape design, vector magnetic properties

## Abstract

In this paper, a stator core shape design method is proposed for an improvement in the power density of a small surface-mounted permanent magnet (SPM) motor. In order to improve the power density of a motor, it is necessary to increase its torque or reduce its weight. However, when a stator core shape is miniaturized to reduce the weight of the motor, the winding regions in a motor frequently decrease. Our stator core shape design method improves the power density of a motor by reducing its weight without decreasing the torque and keeping the winding regions constant. Moreover, the core loss of a motor also decreases when using our method. A Bezier curve is used for the determination of a stator core shape. The finite element method in consideration of the vector magnetic properties is used to evaluate the core loss of the motor shaped by our method. As a result, the power density of an SPM motor is improved, and the core loss of the motor decreases.

## 1. Introduction

Recently, the development of an unmanned space probe has become very important for space exploration. However, the cost for launch becomes large in proportion to the weight of a space probe. To launch a space probe into space by a rocket, the space probe must be as light weight as possible. Since many kinds of small motors are used for a space probe, it is necessary to realize the light weight and small size of a motor without decreasing the torque of a motor. Furthermore, a motor used for a space probe must be efficient in order to utilize a limited energy in space effectively.

Although there have been investigations into the development of the high power density of a motor, they are not investigations about a small motor of less than 30 mm [1,2,3,4,5]. Moreover, it is difficult to miniaturize those high power density motors because those motors have a complex structure. As for a small motor, the structure becomes simple. A surface-mounted permanent magnet (SPM) motor that is used generally for a space probe was selected as a small motor for the development of a high power density motor. A stator core shape in a small SPM motor was investigated for improving power density [6]. When the tooth length is shortened, the power density increases, because the torque increases and the weight of the motor decreases. Moreover, core loss decreases because the weight of the stator core decreases. However, it is necessary to make the winding fill factor increase because the winding region decreases in this case.

On the other hand, it is difficult to dissipate the heat of a small motor used for a space probe because it is used in a vacuum. It is necessary to suppress the generation of the heat in a motor as much as possible. In this paper, the core loss of a small SPM motor increases in order to achieve the high power of a motor by high-frequency excitation. Therefore, we have to decrease the core loss of a motor. Although there have been investigations into the reduction of core loss of a motor, they have carried out the core loss reduction of a stator core by replacing a general steel sheet with nonconventional magnetic materials with low core loss [7,8,9,10]. However, the nonconventional magnetic materials are unsuitable for the stator core of a small motor of less than 30 mm from the viewpoints of difficulty of process. Therefore, the stator core of our small motor is made of a steel sheet. Furthermore, although there have been investigations into the reduction of harmonic core loss, those techniques are not effective as the core loss reduction of our small motor because our motor is driven by a three-phase sinusoidal wave voltage but not pulse width modulation (PWM) [11,12]. Therefore, a stator core shape in a small SPM motor was investigated for core loss reduction [13]. Core loss is reduced by attaching the corner radius R_b_ to the tooth root of the previous stator core shape shown with the broken line, as shown in Figure 1. The core loss reduction effect is large when the corner radius R_b_ is large. However, a winding region decreases because a stator core region increases when the corner radius R_b_ increases, as shown in Figure 1. Consequently, when a sufficient winding region is not securable, it is necessary to make the winding fill factor increase in order to obtain the same value as the back electromotive force (EMF) of a motor with a previously stator core shape. Generally, it is difficult to manufacture a small motor with a high winding fill factor. Additionally, as shown in Figure 1, the area of the stator core increases by attaching the corner radius R_b_ to the tooth root, so that the motor weight also increases. Unless the torque of the motor is increased, the power density of the motor decreases. Therefore, it is necessary to establish a stator core design method that keeps a winding region constant when improving the power density of a small motor.

In this paper, a stator core shape design of a small SPM motor for low core loss, high power density, and keeping a winding region constant is proposed. For the confirmation of low core loss on the stator core shape design, the analysis technique, which can obtain the detailed core loss distribution of a stator core, is required. Although there have been investigations into the calculation of the core loss of a motor, those techniques can obtain total core loss but cannot obtain core loss distributions [14,15,16,17,18]. Furthermore, although core loss distribution is obtained in [19], it is not taking into consideration the rolling direction of the steel sheet. Since the rolling direction of the steel sheet affects the magnetic flux density distribution of a motor, the rolling direction cannot be ignored in the analysis of a small motor [20]. However, we can evaluate the core loss of a motor by changing the shape of a stator core because we have the analysis technique using a detailed magnetic property called the vector magnetic property [21]. The accuracy of our analysis technique has been previously reported [22]. The analysis technique is described in the next section.

## 2. Analysis Condition and Analysis Technique

We previously investigated the characteristics of a motor with the same shape as the base of an analysis model [6,20]. We use again the same motor shape as a base model in this paper. Figure 2 shows the base model in our stator core shape design. In our stator shape design method, only a stator shape is changed without changing a rotor shape. Both the lamination thickness of the stator and the rotor are constant at 12.5 mm. In this paper, we analyze the motor by using the database of vector magnetic properties of the non-oriented steel sheet in which accuracy is already verified. The arrows in the stator core show the rolling direction of the steel sheet. The stator winding method is a concentrated winding. Coil U*, V*, and W* respectively mean the coil of an opposite winding direction to the coil U, V, and W. The magnets used for an SPM rotor are magnetized in parallel with the direction of arrows, and the remanent magnetization of the magnets is constant at 0.4 T.

Generally, the current flowing through a coil is determined by the applied voltage, coil resistance, and coil inductance in a motor. Consequently, the characteristics of a motor, such as the core loss and torque, should be compared under the same applied voltage condition. However, if the stator core shape is changed, it is expected that the value of the back EMF will change because the slot shape is also changed. Therefore, in order to compare and investigate the various motors whose stator core shape is different, analysis conditions of the motors are made consistent according to the following procedure.
Change the number of winding turns to keep the effective value of the back EMF constant whether the stator shape is changed.Calculate the coil resistance using the turn’s number ratio because the diameter of the winding wire is constant.Analyze various motors with different stator core shapes by inputting the same three-phase sinusoidal wave voltage.

It is possible to compare and investigate motors of different structures. In this paper, the voltage effective value of a three-phase sine wave is 6 V and the frequency *f* is 667 Hz (i.e., 10,005 rpm).

In order to evaluate the core loss and torque of the motor designed by our method, the motor is analyzed using the Finite Element Method (FEM) program, which we created by MATLAB. The Enokizono & Soda (E&S) model is introduced in FEM because the E&S model can express in detail a vector magnetic property of an electromagnetic steel sheet used as a stator core [21]. We previously reported the accuracy of our FEM program into which the E&S model was introduced [22]. The vector magnetic properties have the alternating magnetic flux condition and the rotating magnetic flux condition. Figure 3a,b show the alternating magnetic flux condition and the rotating magnetic flux condition, respectively. *B*_max_ is the maximum flux density. The inclination angle *θ_B_* is defined as the angle between the rolling direction and the direction of the maximum flux density vector. The axis ratio *α* is the ratio of the minimum flux density *B*_min_ to the maximum flux density *B*_max_. These three parameters define the flux density condition on the vector magnetic property. Therefore, the E&S model is expressed as a function of these parameters.

The E&S model introduced into FEM is as follows:(1)$$H_{k}\left(\tau \right)=\nu_{kr}\left(B_{\max},\theta_{B},\alpha ,\tau \right)B_{k}\left(\tau \right)+\nu_{ki}\left(B_{\max},\theta_{B},\alpha ,\tau \right)\int B_{k}\left(\tau \right)d\tau ,$$
where subscript *k* = *x* or *y*, *ν_kr_* is the magnetic reluctivity coefficient, and *ν_ki_* is the magnetic hysteresis coefficient [21]. The 2D governing equation in this analysis is as follows:(2)$$\frac{\partial }{\partial x}\left(\nu_{yr}\frac{\partial A}{\partial x}\right)+\frac{\partial }{\partial y}\left(\nu_{xr}\frac{\partial A}{\partial y}\right)+\frac{\partial }{\partial x}\left(\nu_{yi}\int \frac{\partial A}{\partial x}d\tau \right)+\frac{\partial }{\partial y}\left(\nu_{xi}\int \frac{\partial A}{\partial y}d\tau \right)=-J_{0}-\nu_{0}\left(\frac{\partial M_{y}}{\partial x}-\frac{\partial M_{x}}{\partial y}\right),$$
where *A*, *J*_0_, *ν*_0_, *M_x_*, and *M_y_* respectively are the magnetic vector potential, exciting current density, magnetic reluctivity in vacuum, and x and y-components of magnet magnetization. Additionally, the circuit equation is written as
(3)$$V_{0n}=\frac{\partial }{\partial t}\int_{c}Ads+R_{mn}I_{mn},$$
where *V*_0*n*_ (*n* = 1–3), *R_mn_*, and *I_mn_* respectively are the terminal voltage, resistance of an exciting coil, and exciting current. In our analysis, the core loss in each finite element, *P_i_* is calculated directly from analysis results by the following equation:(4)$$P_{i}=\frac{1}{\rho T}\int_{0}^{T}\left(H_{x}\frac{dB_{x}}{dt}+H_{y}\frac{dB_{y}}{dt}\right)dt,$$
where *ρ* is the material density and *T* is the period of the exciting waveform. This analysis technique is able to calculate the core loss distribution in a motor core directly from magnetic flux density vector ***B*** and magnetic field intensity vector ***H*** by using Equation (4). Additionally, the total core loss can be calculated as the sum of core loss *P_i_* [22].

Power density is calculated from the torque and weight of the motor. Torque is calculated from the analyzed result by using Maxwell’s stress tensor method. The weight of the motor is calculated as the sum of the weight of the stator core, coils, and rotor. Since the diameter of the winding wire is constant at 0.38 mm, the weight of the coils is calculated from the number of winding turns. The weight of the rotor is constant at 11 g. The stator core weight is calculated based on the shape of the stator core.

## 3. Stator Core Shape Design Method

In this paper, the stator core shape design for improving the power density of a small motor is carried out in two steps of designs, the “shape design for core loss reduction” and “shape design for improving power density”, in the following sections.

### 3.1. Shape Design for Core Loss Reduction

Although attaching the corner radius to the tooth root as shown in Figure 1 is effective for core loss reduction, a problem in which the winding region decreases occurs. Therefore, the method of smoothing the corner of the tooth root without decreasing the winding region is proposed. A Bezier curve is used for the determination of the stator core shape for smoothing the corner of the tooth root. As shown in Figure 4, a Bezier curve is determined at the four control points P_1_, P_2_, P_3_, and P_4_. The vectors ***P*_12_** and ***P*_34_** determined by P_1_–P_2_ and P_3_–P_4_ are the tangents at the ends P_1_ and P_3_ of the curve, respectively. In this paper, the shape and mesh division of the motor are generated by the software FEMAP. The mesh is divided by a linear triangular element.

Figure 5 shows 1/9th of the stator shape of the base model and Bezier model. The base model in Figure 5a is the same as the stator shape in Figure 2. The Bezier model in Figure 5b is the stator core shape, which smoothed the corner of the tooth root in the base model by a Bezier curve. By adjusting the vector ***P*_12_** and ***P*_34_** of a Bezier curve, the stator core shape of the Bezier model is determined so that the area of the winding region of the Bezier model becomes almost the same as that of the base model. In order to smooth the corner of the tooth root and keep a winding region constant, it is necessary to narrow the width in the middle of the tooth, as shown in Figure 5b. If the tooth width is narrowed, the increase of torque is expected because the flux density in the stator core increases and the flux density in the air gap also increases. Conversely, narrowing the width too much causes the increase of core loss. Therefore, it is necessary to pay attention to the tooth width when the stator shape is determined. As shown in Figure 5, the tooth width of the base model is 2.4 mm uniform, and the tooth width at the narrowest position of the Bezier model is 1.94 mm. Additionally, in this paper, an optimal design was not used to determine the shape of the Bezier model. Since the Bezier model was created by trial and error, it is not the optimal shape.

Table 1 shows the number of turns in phase winding, the coil resistance, and the winding fill factor for the base model and Bezier model. In this investigation, the number of winding turns is not a whole number, because the number of winding turns is changed to keep the effective value of the back EMF constant. Therefore, the winding coil with turn numbers shown in Table 1 cannot be manufactured actually because the number of winding turns is not whole numbers. In the Bezier model, since the effective value of the back EMF slightly decreases as compared with the base model, the number of winding turns is increased slightly. However, since the winding regions of the Bezier model become bigger after modification, the winding fill factor of the Bezier model is reduced 0.5% from that of the base model.

Next, we compare the weights of the base model and Bezier model. Table 2 shows the weight of the coils, stator core, and motor for the base model and Bezier model. The coil’s weight in the Bezier model increases 0.4% because the number of winding turns increases as compared with the base model. On the other hand, the weight of the stator core in the Bezier model decreases about 1% from that of the base model. Consequently, the weight of the motor of the Bezier model decreases 0.5% from that of the base model.

Figure 6 shows the core loss distribution of the base model and Bezier model. As shown in Figure 6b, since magnetic flux density slightly increases by narrowing the tooth width, the local core loss in the tooth slightly also increases. On the other hand, the local core loss in the back yoke region decreases. This is because the corners of the tooth root region are smoothed by the Bezier curve, and the magnetic flux concentrations to the corners are suppressed.

Figure 7 shows the core loss for the torque of the base model and Bezier model. The core loss is calculated as the sum of core loss distribution shown in Figure 6. As shown in Figure 7, the effectiveness of core loss reduction in the Bezier model is confirmed. The core loss of the Bezier model decreases about 5% from that of the base model by effect of the local core loss reduction in the back yoke.

Figure 8 shows the power density for the load angle of the base model and Bezier model. Since the tooth width is narrowed and flux density in the stator core increases, the torque of the Bezier model increases. Additionally, as shown in Table 2, the motor weight of the Bezier model is lighter than that of the base model. Therefore, the power density of the Bezier model is larger than that of the base model.

### 3.2. Shape Design for Improving Power Density

Next, in order to improve the power density of the Bezier model without decreasing the torque, the stator core design by processing of the Bezier model is proposed.

Figure 9 shows the magnetic flux density distribution of the base model and Bezier model. These figures are shown by the low magnetic flux density range in order to emphasize the region of low magnetic flux density. As compared with the base model, the Bezier model has a lower magnetic flux density in the back yoke part surrounded by the broken line. This is because the corners of the tooth root region are smoothed, and the magnetic fluxes around the corners do not protrude to the back yoke side. Even if these low magnetic flux density regions in the back yoke are cut out, they hardly affect the magnetic circuit, and it is considered not to lead to a reduction in torque. As a result, it is possible to reduce only the stator core weight and improve the power density. Although the base model has also low magnetic flux density regions in the back yoke part, the regions are not large compared with the Bezier model. Therefore, even if the regions of the base model are cut out, an improvement of the power density of the base model is not expected so much.

A region of the back yoke that is cut out in the Bezier model is shown in Figure 10. Figure 10 is an enlarged view of the back yoke portion surrounded by the broken line in Figure 9b. In this paper, the range less than 0.2 T or 0.3 T is considered to be the low magnetic flux density range. Therefore, the region to cut out is set to two circles with radiuses of 1.25 mm and 1.50 mm in the low magnetic flux density range.

Figure 11a shows the R1.25 model, which is the Bezier model with the back yoke cut out by arcs with a radius of 1.25 mm. Figure 11b shows the R1.50 model, which is the Bezier model with the back yoke cut out by arcs with a radius of 1.50 mm. Since edges occur at the back yoke after being cut out, a corner radius of 1.00 mm is attached to the edges in order to reduce the core loss, as shown in Figure 11a,b.

Table 3 shows the weight of the coils, stator core, and motor for the R1.25 model and R1.50 model. Since the back EMF is not affected even if the low magnetic flux density region in the back yoke is cut out, the number of winding turns does not change. Therefore, the coil weights in both models are the same as that of the Bezier model. Obviously, the stator core weights of both models are lighter than that of the Bezier model. Accordingly, the decreases of motor weight in the R1.25 model and R1.50 model from the base model are about 5% and 7%, respectively.

Figure 12 shows the core loss distribution of the Bezier model and R1.25 model. The core loss distribution in the tooth of the R1.25 model is almost the same as that of the Bezier model. However, in the R1.25 model, the local core loss around the region cut out in the back yoke increases slightly. As shown in Figure 9, the shapes of the low magnetic flux density region in the back yoke are distorted under the influence of rolling direction and are not completely rotational symmetry [20]. In this paper, the rotational asymmetry of magnetic flux density distribution was not taken into consideration when cutting out the low magnetic flux density regions. Therefore, since the low magnetic flux density regions were cut out by the same shape at all nine points in the back yoke, it is assumed that the core loss increased.

Figure 13 shows the core loss for the torque of the Bezier model, R1.25 model, and R1.50 model. As shown in Figure 13, the core losses in both models slightly increase under the influence of the cut-out yoke. The increases of core loss in the R1.25 model and R1.50 model from that of the Bezier model are 1.3% and 2.6%, respectively. This reason is because the width of the magnetic path narrowed under the influence of the cut-out region, and iron loss increased at the places surrounded by red circles shown in Figure 14b. Therefore, we need to pay attention to the range of the region to cut out.

Figure 15 shows a comparison of the torque waveforms in the Bezier model, R1.25 model, and R1.50 model. As shown in Figure 15, all of the torque waveforms of the three models are the same. This result indicates that cutting out the low magnetic flux density regions in the back yoke does not affect torque.

Figure 16 shows the power density for the load angle of the base model, R1.25 model, and R1.50 model. As shown in Figure 15, the torques of the R1.25 model and R1.50 model are the same as that of the Bezier model, and as shown in Table 3, the motor weights of the R1.25 model and R1.50 model are lighter than that of the Bezier model. Therefore, the power densities of the R1.25 model and R1.50 model are larger than that of the Bezier model. As a result, as shown in Figure 16, those power densities are greatly improved compared with the base model. The increases of power density in the R1.25 model and R1.50 model from the base model are about 9% and 12%, respectively.

## 4. Conclusions

In this paper, the stator core shape design for improving the power density of a small SPM motor is proposed. The design is carried out in two steps. The first design reduces the core loss, and the second design improves the power density. The Bezier curve used for the first design not only decreases the core loss of a motor, but it also has an effect that extends the low magnetic flux density region in the back yoke part. Therefore, in the second design, the power density of a motor is improved by cutting out the low magnetic flux density region, without decreasing the torque. As a result, it is shown that our technique is useful as a stator core shape design method for decreasing core loss and improving power density. However, neither the stator core shape defined by a Bezier curve nor the shape cut out in the back yoke have been optimized. The characteristics of the small motor may become better by an optimal design. Additionally, since these results are theoretical, it is necessary to actually manufacture a small SPM motor and verify the validity of these results.

## Figures and Tables

**Figure 1 sensors-20-01418-f001:**
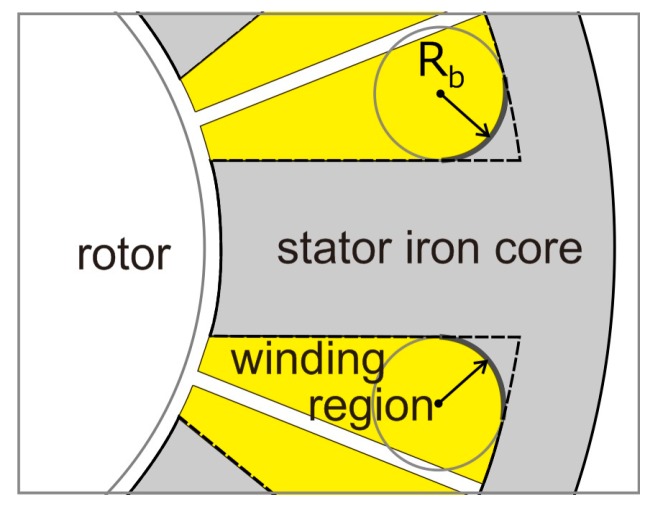
Core loss reduction by attaching the corner radius R_b_ to the tooth root.

**Figure 2 sensors-20-01418-f002:**
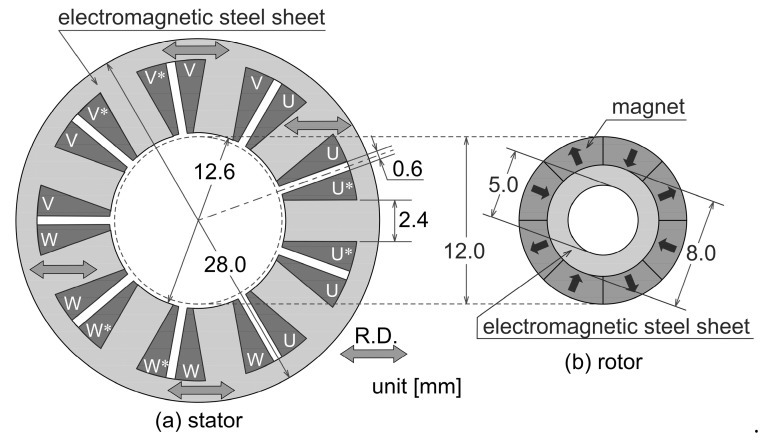
The base model in the stator core shape design.

**Figure 3 sensors-20-01418-f003:**
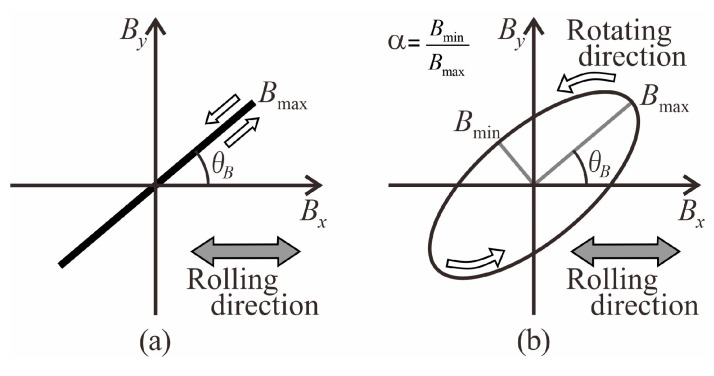
Definition of (**a**) alternating magnetic flux condition and (**b**) rotating magnetic flux condition.

**Figure 4 sensors-20-01418-f004:**
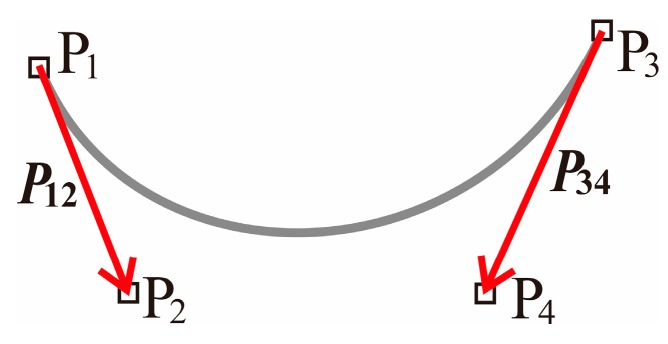
Definition of a Bezier curve.

**Figure 5 sensors-20-01418-f005:**
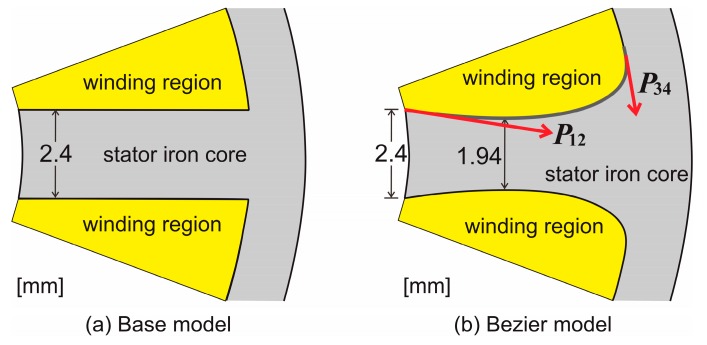
Stator core shape and winding region of (**a**) the base model and (**b**) Bezier model.

**Figure 6 sensors-20-01418-f006:**
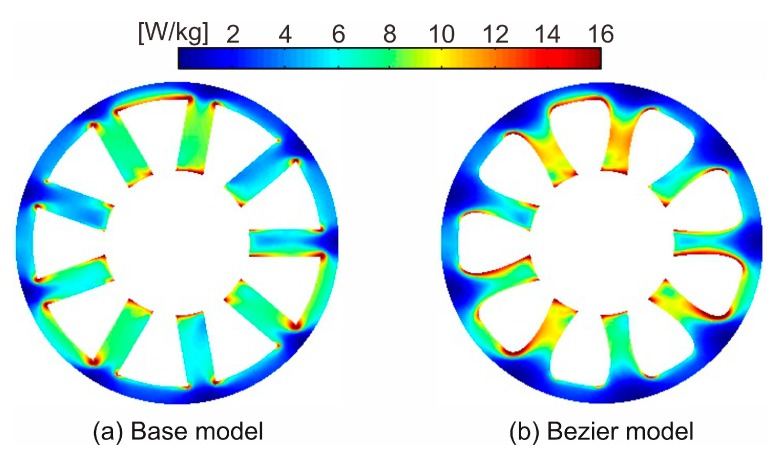
Core loss distribution of (**a**) the base model and (**b**) Bezier model.

**Figure 7 sensors-20-01418-f007:**
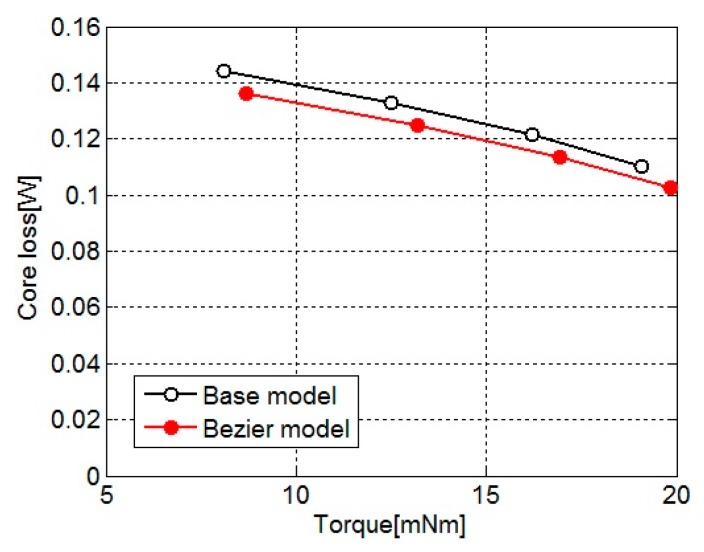
Core loss for torque of the base model and Bezier model.

**Figure 8 sensors-20-01418-f008:**
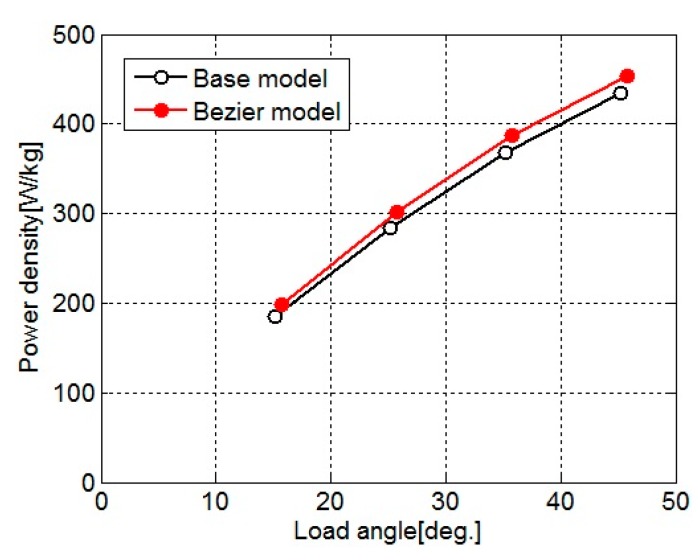
Power density for the load angle of the base model and Bezier model.

**Figure 9 sensors-20-01418-f009:**
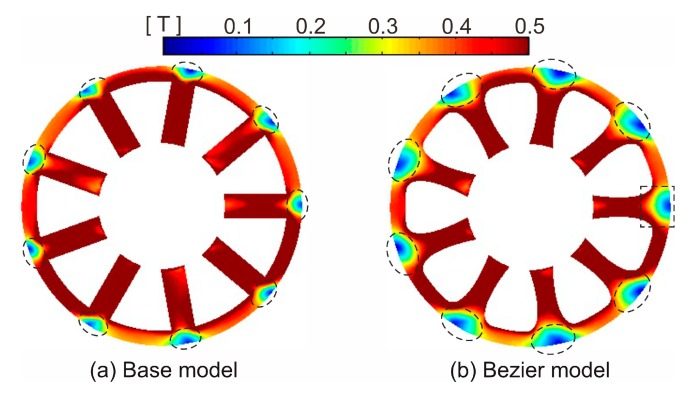
Magnetic flux density distribution of (**a**) the base model and (**b**) Bezier model shown by the low magnetic flux density range.

**Figure 10 sensors-20-01418-f010:**
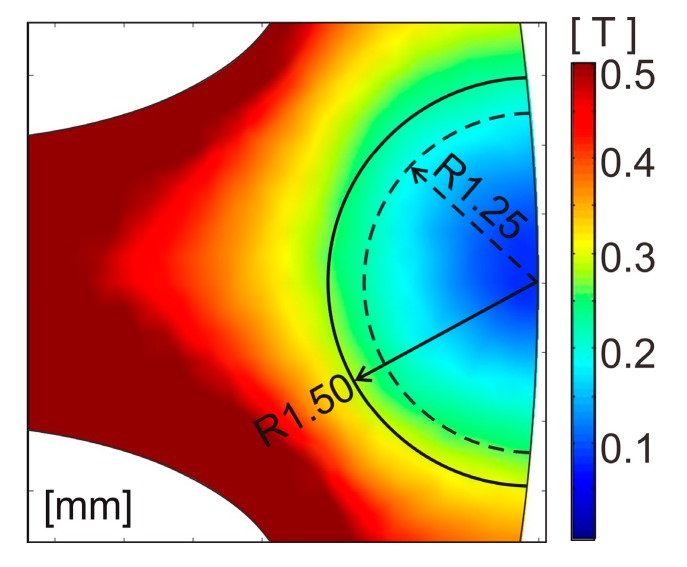
Enlarged view of a region of the back yoke, which is cut out in the Bezier model.

**Figure 11 sensors-20-01418-f011:**
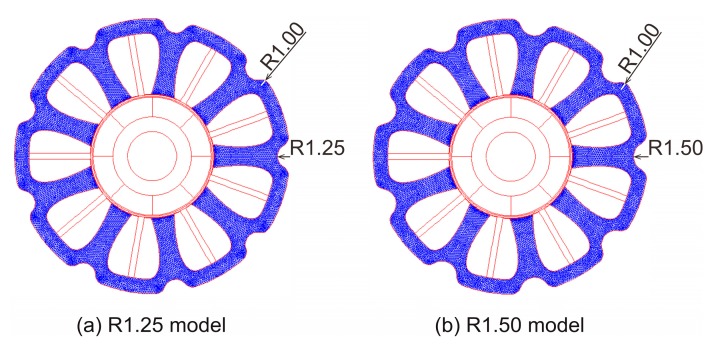
(**a**) R1.25 model and (**b**) R1.50 model, which are the Bezier model with the back yoke cut out by arcs with radiuses of 1.25 mm and 1.50 mm, respectively.

**Figure 12 sensors-20-01418-f012:**
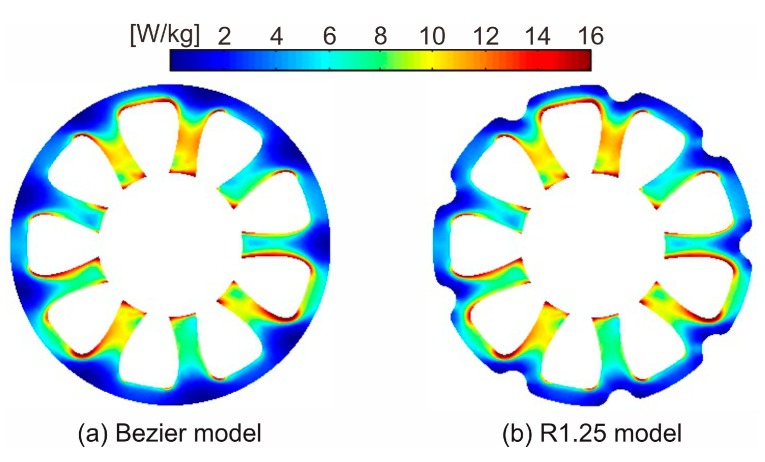
Core loss distribution of (**a**) the Bezier model and (**b**) R1.25 model.

**Figure 13 sensors-20-01418-f013:**
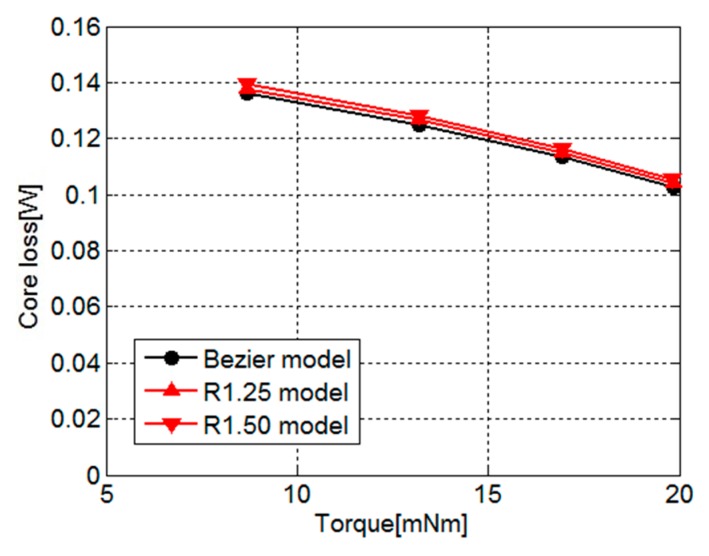
Core loss for torque of the Bezier model, R1.25 model, and R1.50 model.

**Figure 14 sensors-20-01418-f014:**
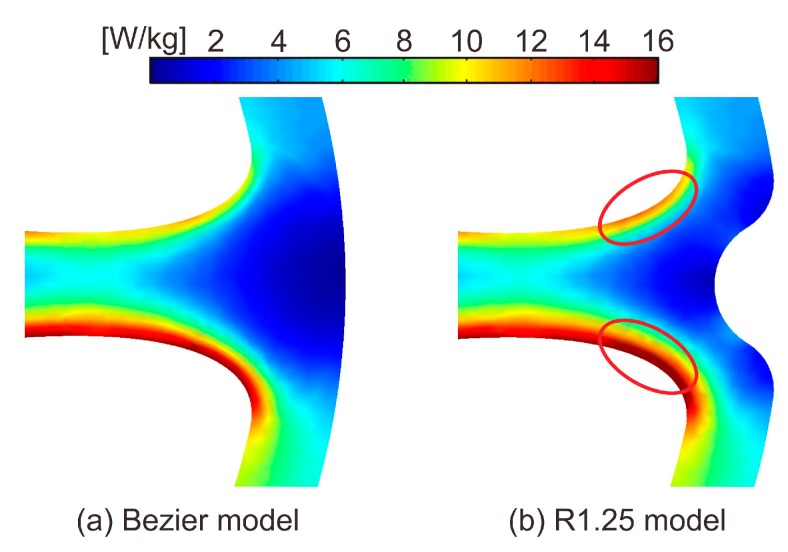
Enlarged view of a region of the back yoke in core loss distribution of (**a**) the Bezier model and (**b**) R1.25 model.

**Figure 15 sensors-20-01418-f015:**
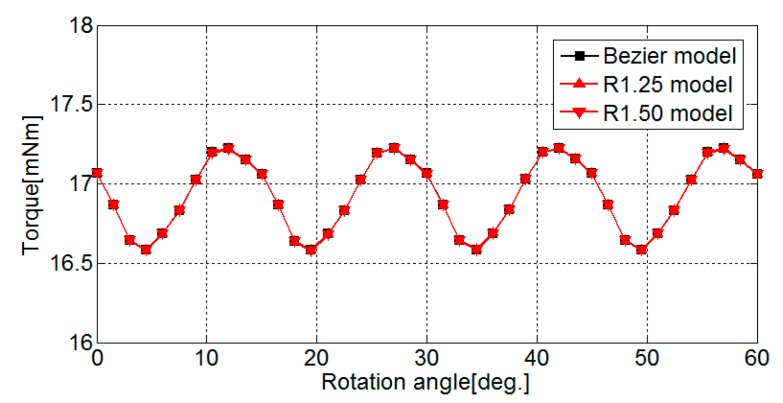
Comparison of torque waveforms in the Bezier model, R1.25 model, and R1.50 model.

**Figure 16 sensors-20-01418-f016:**
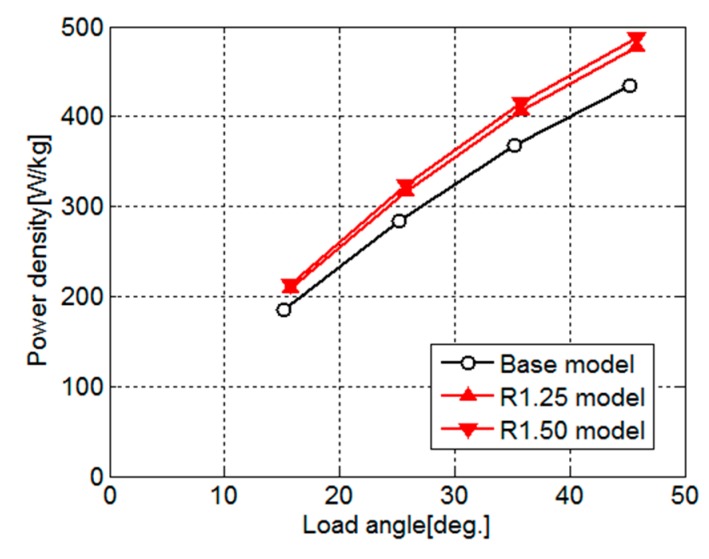
Power density for load angle of the base model, R1.25 model, and R1.50 model.

**Table 1 sensors-20-01418-t001:** Number of turns in phase winding, coil resistance, and winding fill factor for the base model and Bezier model.

Model	Number of Turns in Phase Winding	Coil Resistance [Ω]	Winding Fill Factor [%]
Base model	44.71	0.9853	46.02
Bezier model	44.87	0.9888	45.54

**Table 2 sensors-20-01418-t002:** Weight of coils, stator core, and motor for the base model and Bezier model.

Model	Coil’s Weight [g]	Stator Core’s Weight [g]	Motor’s Weight [g]
Base model	10.15	24.95	46.10
Bezier model	10.19	24.68	45.87

**Table 3 sensors-20-01418-t003:** Weight of coils, stator core, and motor for the R1.25 model and R1.50 model.

Model	Coil Weight [g]	Stator Core Weight [g]	Motor Weight [g]
R1.25 model	10.19	22.41	43.60
R1.50 model	10.19	21.48	42.67

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
