# Peer review of "Stator Core Shape Design for Low Core Loss and High Power Density of a Small Surface-Mounted Permanent Motor"

_sensors, 2020, doi:10.3390/s20051418_

Round 1

Reviewer 1 Report

This paper suggested a new stator core shape design for better efficiency. However, In the introduction section; the literature review, scientific discussion and need for a new design were not reported clearly. The Finite Element Model which was taken from previous references ( Ref 4) would be more described. 

Below issues should be justified in the paper; 

Which Finite Element software did you use for the analysis? If it's a commercial one, you should cite it.  The equations of the Bezier curve would be included, there was a lack of information about how you decided this curve...  Some concerns on the results; How did you validate your Finite Element results? Did you make any convergence studies for your analyses? Which element did you choose for your model? which material properties (steel sheet data from NIPPON KINZOKU [3], is it enough for modelling?), did you use for the analysis ? are your analyses repeatable?, etc.   Why did you choose R.125 and R1.50 model, how did you decide these radii for comparison? Analyses can be expanded with more pertinent parameters.

Author Response

Dear Reviewer,

Thank you for your suggestion.

Q:In the introduction section; the literature review, scientific discussion and need for a new design were not reported clearly.

Ans:We added the lines 30-34 and the references 1-4 to my article.

Q:The Finite Element Model which was taken from previous references ( Ref 4) would be more described.

Ans:We added the lines 90-97, Figure 3, and the lines 102-106 to my article.

Q:Which Finite Element software did you use for the analysis? If it's a commercial one, you should cite it. 

Ans:We added the lines 87-88 to my article.

Q:The equations of the Bezier curve would be included, there was a lack of information about how you decided this curve...

Ans:We added the lines 134-135 and the lines 149-150 to my article.

Q:Some concerns on the results; How did you validate your Finite Element results? Did you make any convergence studies for your analyses?

Ans:We added the lines 89-90 to my article.

Q:Which element did you choose for your model?

Ans:We added the line 135 to my article.

Q:which material properties (steel sheet data from NIPPON KINZOKU [3], is it enough for modelling?), did you use for the analysis ? are your analyses repeatable?, etc.

Ans:We added the lines 64-65 to my article.

Q:Why did you choose R.125 and R1.50 model, how did you decide these radii for comparison?

Ans:We already write the reason at the lines 192-193 and the lines 202-204.

Q:Analyses can be expanded with more pertinent parameters. 

Ans:We added the lines 230-232 to my article.

Reviewer 2 Report

All my comments and remarks are discussed in additional files.

Author Response

Dear Reviewer,

Thank you for your suggestion.

Ans1:We corrected as your suggestion.

Ans2:We corrected our title of paper as your suggestion.

Ans3:We added the lines 30-34 and the references 1-4 to my article.

Ans4:We added the references 1-4 to my article.

Ans5:We added the lines 60-61 as a explanation.

Ans6:Purpose of this article is different from previous two article. In “Stator Shape Design Method for Improving Power Density in PM Motor”, motor size is changed. Specifically, teeth length, teeth width and back yoke size were changed. However, in this article, only stator shape is changed but outer diameter is same. Purpose of  “Relation Between Stator Core Shape and Torque Ripple for SPM Motor” is reduction of torque ripple but not improvement of power density.

Ans7:We corrected as your suggestion. Please check the line 119.

Ans8:We corrected "number of winding" to "turn's number in phase winding."

Ans9:We added the line 162 as a explanation.

Ans10:We added the lines 230-232 and Figure 14 as a explanation.

Ans11:We added the lines 194-197 as a explanation.

Round 2

Reviewer 1 Report

Unfortunately, I didn't see any successful progress for the asked issues.   

Author Response

Unfortunately, I didn't see any successful progress for the asked issues. 

Ans: Sorry, I could not fulfill your request, in previous revised paper.

In this revision, we did our best.

This paper suggested a new stator core shape design for better efficiency. However, In the introduction section; the literature review, scientific discussion and need for a new design were not reported clearly. The Finite Element Model which was taken from previous references ( Ref 4) would be more described.

Ans:We increased list of references and added explanation of the research background in the introduction. We added the sentence for explanation at lines 41-50 and lines 63-73, and added the references No. 2, 7-12, 14-19.

Which Finite Element software did you use for the analysis? If it's a commercial one, you should cite it. 

Ans:We didn't use Finite Element software. So, we added the sentence for explanation at lines 102-103: “In order to evaluate core loss and torque of the motor designed by our method, the motor is analyzed using the Finite Element Method (FEM) program which we created by MATLAB.”

The equations of the Bezier curve would be included, there was a lack of information about how you decided this curve... 

Ans:We don’t use equations as definition of the Bezier curve. So, we added the sentence for explanation at lines 145-146 and lines 159-161: “In this paper, the shape and mesh division of motor are generated by the software FEMAP.” and “Additionally, in this paper, an optimal design was not used to determine the shape of Bezier model. Since the Bezier model was created by trial and error, it's not the optimal shape.”

Some concerns on the results; How did you validate your Finite Element results? Did you make any convergence studies for your analyses?

Ans:We added the sentence for explanation at lines 103-107: “E&S model is introduced in FEM because the E&S model can express in detail a vector magnetic property of an electromagnetic steel sheet used as a stator core [21]. We previously reported the accuracy of our FEM program into which the E&S model was introduced [22]. The vector magnetic properties have the alternating magnetic flux condition and the rotating magnetic flux condition.”

Which element did you choose for your model?

Ans: We added the sentence for explanation at lines 145-146: “In this paper, the shape and mesh division of motor are generated by the software FEMAP. The mesh is divided by a linear triangular element.”

Which material properties (steel sheet data from NIPPON KINZOKU [3], is it enough for modelling?), did you use for the analysis? are your analyses repeatable?, etc.  

Ans:We corrected the sentence at lines 80-81: “In this paper, we analyze the motor by using the database of vector magnetic properties of the non-oriented steel sheet in which accuracy is already verified.”

Why did you choose R.125 and R1.50 model, how did you decide these radii for comparison? 

Ans: I already wrote the reason at the lines 212-215: “In this paper, the range less than 0.2 T or 0.3 T is considered to be low magnetic flux density range. Therefore, the region to cut out is set to two circles with the radius of 1.25 mm and 1.50 mm in the low magnetic flux density range.”

Analyses can be expanded with more pertinent parameters.

Ans: I added the sentence in conclusion at lines 269-271: “However, either the stator core shape defined by a Bezier curve and the shape cut out in the back yoke have not optimized. The characteristics of the small motor may become better by an optimal design.”

Reviewer 2 Report

The paper entitled "Stator core shape design for low core loss and high power density of a small SPM motor" presented improved design of small-dimension SPM motor. The design process of studied motor has been carried out in two independent stages. In the first stage of designing process the authors reduce core losses in motor. After this the second stage concerns improvement of the power density in the construction.

In my opinion, the authors tried to partially respond to the comments I indicated in the first review. All grammar and letter errors indicated were corrected. However, I must again refer to the literature prepared by the authors. Despite my comments about quoting only one author's. As I noticed, 5 new articles were added in the improved version. But the problem of literature is still actual, because out of nine articles in the list of references, five are from the same author. This gives about 55% of the articles of one of the authors from reference list. Despite the previous my request, sufficient corrections have not been made in this respect.

Some valuable information has been added to Chapter 2, "Analysis Condition and Analysis Technique." For me valuable information is about developed in-house FEM program in Matlab. In the second chapter, a mathematical model was built, but the authors, instead of improving this fragment create it more complicated for readers. In this chapter, I am not satisfied with the level of corrections.

Below are further comments for the authors:

  1. The references are not numbered according to their appearance in the text.
  2. Row 72. What mean "inductance of the motor".
  3. In the Table 1, authors provide information that, the turn’s number in phase winding is 44.71. So how wound in the slots the non-integer number of turns?
  4. The quality of Figure.15 is insufficient. The author compares waveforms for three different variants. These waveforms overlap each other.
  5. The article summary should also be improved.

Author Response

In my opinion, the authors tried to partially respond to the comments I indicated in the first review. All grammar and letter errors indicated were corrected. However, I must again refer to the literature prepared by the authors. Despite my comments about quoting only one author's. As I noticed, 5 new articles were added in the improved version. But the problem of literature is still actual, because out of nine articles in the list of references, five are from the same author. This gives about 55% of the articles of one of the authors from reference list. Despite the previous my request, sufficient corrections have not been made in this respect.

Ans: Sorry, I could not fulfill your request, in previous revised paper. In this revision, we did our best. We increased list of references and added explanation of the research background in the introduction. We added the sentence for explanation at lines 41-50 and lines 63-73, and added the references No. 2, 7-12, 14-19.

Some valuable information has been added to Chapter 2, "Analysis Condition and Analysis Technique." For me valuable information is about developed in-house FEM program in Matlab. In the second chapter, a mathematical model was built, but the authors, instead of improving this fragment create it more complicated for readers. In this chapter, I am not satisfied with the level of corrections.

Ans: Sorry, we mistook how to correct our paper. So, we deleted the complex equations, and quoted literature like previous paper.

Below are further comments for the authors:

1. The references are not numbered according to their appearance in the text.

Ans: Sorry, we increased list of references and numbered according to their appearance.

2. Row 72. What mean "inductance of the motor".

Ans: We corrected the sentence at lines 87-88: “Generally, current flowing through a coil is determined by applied voltage, coil resistance, and coil inductance in a motor.”

3. In the Table 1, authors provide information that, the turn’s number in phase winding is 44.71. So how wound in the slots the non-integer number of turns?

Ans: We added the sentence for explanation at lines 164-167: “In this investigation, the turn's number of winding is not whole number because the turn's number of winding is changed to keep the effective value of back EMF constant. Therefore, winding coil with turn's number of Table 1 cannot be manufactured actually because turn's number of winding is not whole number.”

4. The quality of Figure.15 is insufficient. The author compares waveforms for three different variants. These waveforms overlap each other.

Ans: Since the wave is almost in agreement, these waveforms overlap each other. And We wrote the sentence for explanation at lines 250-251: “As shown in Figure 15, all of torque waveforms of three models are the same. This result indicates that cutting out the low magnetic flux density regions in back yoke does not affect torque.” So, how should we correct Figure 15?

5. The article summary should also be improved.

Ans: We added the sentence for summary at lines 269-272: “However, either the stator core shape defined by a Bezier curve and the shape cut out in the back yoke have not optimized. The characteristics of the small motor may become better by an optimal design. Additionally, since these results are theoretical, it is necessary to actually manufacture a small SPM motor and verify the validity of these results.”

Round 3

Reviewer 2 Report

Thank you very much for the authors' answers to my questions.
I accept all these answers. I have no more comments at this stage.

However, I am most surprised that the number of turns in the winding coil is not integer number. I can not built a prototype with such number of coils.